# Different Amounts of Water Supplementation Improved Cognitive Performance and Mood among Young Adults after 12 h Water Restriction in Baoding, China: A Randomized Controlled Trial (RCT)

**DOI:** 10.3390/ijerph17217792

**Published:** 2020-10-24

**Authors:** Jianfen Zhang, Na Zhang, Hairong He, Songming Du, Guansheng Ma

**Affiliations:** 1Department of Nutrition and Food Hygiene, School of Public Health, Peking University, Beijing 100191, China; ZJF@bjmu.edu.cn (J.Z.); ziqingxuanping@126.com (N.Z.); hhrhhr3@163.com (H.H.); 2Laboratory of Toxicological Research and Risk Assessment for Food Safety, Peking University, Beijing 100191, China; 3Chinese Nutrition Society, Beijing 100053, China; dusm9709@126.com

**Keywords:** water restriction, dehydration, water supplementation, rehydration, cognitive performance, mood

## Abstract

Water is indispensable to keeping the functions of the human body working properly, including that of the brain. The purpose of this research was to explore the impacts of water supplementation on cognitive performance and mood, and to determine the optimum amount of water to alleviate detriments of dehydration after 12 h water restriction. A randomized controlled trial was implemented among 64 young adults from Baoding, China. Fasting overnight for 12 h, and at 8:00 a.m. on day 2, osmolality of first morning urine and blood, cognitive performance, and mood were assessed as the dehydration test. Then, participants were randomly separated into four groups: water supplementation groups (WS groups 1, 2, and 3 with 500, 200, and 100 mL purified water, respectively) and no water supplementation group (NW group). Participants in WS groups were instructed to drink the water within 10 min, while those in NW group drank no water. After 90 min, the same measurements were taken as the rehydration test. There was significant interaction between TIME and VOLUME in thirst when comparing dehydration with rehydration tests (*F* = 6.172, *p* = 0.001). Significant thirst reductions were found in WS group 1 and WS group 2 (*p* = 0.003; *p* = 0.041), and a significant increase was found in the NW group (*p* = 0.039). In the rehydration test, significant interactions between TIME and VOLUME were found in scores of anger, fatigue, and TMD (total mood disturbance) (*F* = 3.815, *p* = 0.014; *F* = 10.429, *p* < 0.001; *F* = 5.246, *p* < 0.001), compared to the dehydration test. Scores of anger were only decreased in WS group 2 (*p* = 0.025), and scores of fatigue and TMD decreased in WS group 1 and WS group 2 (all *p* < 0.05). Significant interaction between TIME and VOLUME was only found for operation span test scores (*F* = 2.816, *p* = 0.047), with scores being only higher in WS group 1 in the rehydration test compared to the dehydration test (*p* = 0.003). Comparing WS group 1 and WS group 2, scores of thirst, fatigue, and TMD did not differ significantly (*p* > 0.05). Water supplementation improved working memory and attenuated anger, fatigue, and TMD. A small amount of water (200 mL) was sufficient to attenuate thirst, anger, fatigue, and TMD of young adults, but the larger volume (500 mL) appeared to be necessary to improve working memory. The amount of 500 mL was the optimum volume to improve the cognitive performance and mood among young adults.

## 1. Background

Water is the basic element of the human body, composing 60%–70% of body weight [1,2]. It is crucial for every system and metabolic activity of the human body [3]. Generally, water is in a dynamic balance, with intake equal to the output. Adequate intake of water is recommended by different countries according to the characteristics of the people [4,5], in order to sustain the functions of the body properly.

Despite the importance of adequate water intake to maintain optimal hydration in the human body, studies have shown that adults, including the elderly people, and children do not drink enough water, which may increase the risk of dehydration [6,7,8]. Dehydration is the process of water loss in the human body. The relationship between dehydration and physical performance has been evaluated, showing that dehydration has a negative effect on muscle endurance, power, and strength [9,10]. Moreover, the impacts of dehydration on cognitive performance have also been explored among people of different ages, including children, young adults, and the elderly, in experimental settings [11,12,13,14,15]. Among young adults, exercise or heat stress induced dehydration has detrimental impacts on cognitive performance and mood, including negatively affecting the working memory and attention span, and aspects of mood including TMD (total mood disturbance), fatigue, confusion, anger, and depression are also increased [16,17,18]. Nevertheless, studies showed that exercise and heat stress may have effects on cognitive performance [19,20,21,22]. Therefore, it is necessary to conduct research on water restriction, only to induce dehydration, in order to investigate the impacts of dehydration on cognitive performance and mood (after adjusting for confounding factors). A smaller body of studies has explored the impact of dehydration after water or fluid deprivation on cognitive performance and mood. Furthermore, in China, only one study has been implemented to explore the association between hydration status and cognitive performance of adults [23], which remains an inadequately studied topic.

Regarding the effects of water supplementation intervention on cognitive performance and mood, conflicting findings also exist. Some results support the hypothesis that cognitive performance and mood were improved after water supplementation. A study conducted among young adults revealed that the RVIP task (rapid visual information processing task) improved after water consumption of 120 and 300 mL [24]. Furthermore, results of a study implemented among 47 adults showed that water consumption affected performance in the letter cancellation task, which investigated the cognitive ability of visual attention, but not mood [25]. Similarly, studies conducted among children reported that even under conditions of mild dehydration, consuming a range from 300 to 1000 mL of water resulted in better cognitive performance [26,27,28]. However, some studies have obtained controversial conclusions. It was reported that the impact of mild hypohydration induced by exercise heat stress on the cognitive performance of young adults was not influenced by fluid replacement [29]. Moreover, the results of a study employed among adults indicated that no mood changes were found after participants drank 500 mL water [30]. A randomized controlled trial showed that water supplementation did not impact cognitive performance among primary school pupils living in a hot climate [31]. In sum, whether water supplementation can improve cognitive performance and mood needs to be explored further.

In China, limited research had been conducted in the context of water or hydration status and cognitive performance. Only one study exploring the impacts of water supplementation on cognitive performance and mood has been employed among young male adults, not including women [23]. More studies are needed. Moreover, it remains unclear whether young adults improve more of their cognitive performance, after water restriction for 12 h, by drinking more water. Whilst, the amount of water that is most appropriate to improve cognitive performance and mood is unknown.

Therefore, the objective of this study was to assess the impact of water supplementation on cognitive performance and mood among young adults after fasting for 12 h. In addition, the most appropriate amount of water that attenuated the impairments in cognitive performance and mood induced by dehydration was determined. Furthermore, this research provides implications for policymaking and plays an important role in water education and determining the recommended intake of water in China.

## 2. Methods

### 2.1. Participants

Sixty-four healthy college students, half of them men and half of them women, were recruited from Hebei University Health Science Center.

The inclusion criteria were good health and age 18–23 years. The exclusion criteria were tobacco use, habitual consumption of alcohol (>20 g/day) or caffeine (>250 mg/day), high-intensity exercise habits, nervous system diseases, or other diseases [32].

### 2.2. Sample Size Calculation

A study conducted among young adults demonstrated that [33], the scores of letter cancellation among participants in groups before and after water supplementation were 23.34 and 28.81, respectively. SAS 9.2 (SAS Institute Inc., Cary, NC, USA) was used to calculate the sample size. The power was set at 0.8, α set at 0.05, and a 10% drop-out rate was taken into account. A total of 64 participants were needed.

### 2.3. Study Design and Procedure

The study was a randomized controlled, double blinded trial study design, which was employed among young adults. The study design is demonstrated in Figure 1.

The study was conducted over two days. The indexes of cognitive performance and mood were measured twice during the two days.

Day 1: Water restriction started at 8:00 p.m. on day 1 and ended at 8:00 a.m. on day 2. To fast overnight for 12 h, participants were asked not to have any foods or drinks. Before water restriction, which started at 8:00 p.m. on day 1, participants were allowed to consume food and drink normally. Further, they were instructed that being in bed by 11:00 p.m. was ideal, and they were not allowed to urinate until they got up the next morning.

Day 2: Water supplementation began from 8:00 a.m. to 10:00 a.m. on day 2. At 8:00 a.m., the dehydration test was conducted following standardized procedures. Anthropometric measurements were taken by trained investigators, which included height, weight, and blood pressure. On day 2, participants were instructed to collect their first urine samples using a container designed by the researchers, and the samples were sent to the laboratory to test the osmolality, urinary specific gravity (USG), and concentrations of electrolytes. Venous blood samples were taken by experienced nurses to determine the osmolality, glucose, and electrolyte concentrations. In addition, the visual analog scale (VAS) was used to measure the feeling of thirst, the profile of mood state (POMS) was used to test mood, and software was used to determine cognitive performance. Then, participants were randomly divided into four groups: water supplementation groups (WS groups 1, 2, and 3) and no water supplementation group (NW group). At 8:30 a.m., participants in WS groups 1, 2, and 3 drank, respectively, 500, 200, and 100 mL of purified water according to the instructions of investigators from 8:30 a.m. to 8:40 a.m. Participants in the NW group were not allowed to have any fluids. All water supplied to the participants was put into three opaque cups and maintained at 30–40 °C. During an interval of 90 min (8:30 a.m. to 10:00 a.m.), all participants were only allowed to do light physical activities. At 10:00 a.m., the indexes mentioned above were remeasured in the rehydration test. The study procedure is shown in Figure 2.

In China, people are used to drinking warm or boiled water instead of cold water. Participants would feel uncomfortable if they drank cool or cold water, which may influence the results of the study. Therefore, we maintained the temperature of the water between 30 °C and 40 °C, which is suitable for drinking (neither too hot nor too cold), so as to limit any gastrointestinal upset. The water supplied to the participants was kept in three cups. In order to maintain the temperature of the water, the cups were kept in a water bath, and the researchers measured the temperature of the water periodically during the study.

SPSS software (IBM Corp., Armonk, NY, USA) was used to generate random numbers, and then the subjects were randomly divided into the four groups. In order to make the experiment double-blinded, the researchers who distributed the water to the participants and the participants themselves did not know the amount of water they were supplied. Furthermore, the researchers responsible for data analysis did not know the amounts of water supplied to the groups. During analysis, the groups were named as a, b, c, and d instead of WS group 1, WS group 2, WS group 3, and NW group, respectively. Moreover, every participant was assigned three opaque cups, and the water supplied to them was divided equally into the three cups. They could not know the amount of water in the cups, even from their weight, so they did not know the amount of water in the cups. Further, they were not allowed to talk with each other about the experiment during the study.

### 2.4. Ethical Standards

The study protocol was approved by the Peking University Institutional Review Committee (No. IRB00001052-16071). The study was performed according to the guidelines of the Declaration of Helsinki. All participants signed informed consent forms prior to the study.

### 2.5. Anthropometric Measurements

Height (H) and weight (W) were measured twice using a height and weight meter by an experienced investigator, and participants were asked to wear light clothing and no footwear (HDM-300; Huaju, Zhejiang, China). (BMI: weight (kg)/height squared (m)). Blood pressure (BP) was tested twice by a nurse using an electronic sphygmomanometer (HEM-7051; Omrom, Jiangsu, China). Before measuring the blood pressure, participants were instructed to rest for at least 15 min, and the two measurements were taken with an interval of two minutes. The height was tested twice to the nearest 0.1 cm, and the weight was measured twice to the nearest 0.1 kg. Blood pressure was measured to the nearest 2 mmHg.

### 2.6. Temperature and Humidity of the Environment

The temperature and humidity were measured three times each day, at 10:00 a.m., 2:00 p.m., and 8:00 p.m., including indoors and outdoors, using a temperature hygrometer (WSB-1-H2, Exasace, Zhengzhou, China).

### 2.7. Urinary Biomarker Measurements

Urine was collected from participants using portable plastic urine bags, which were designed by the researchers. After receiving the samples of urine, researchers sent them to the laboratory as soon as possible. Urine samples were analyzed within two hours and stored at +4 °C before taking measurements. Urine osmolality was tested using the freezing point method by an experienced laboratory technician with an osmotic pressure molar concentration meter (SMC 30C; Tianhe, Tianjin, China). The urine electrolyte concentrations were tested by an experienced laboratory technician with an automatic biochemical analyzer (AU 5800; Beckman; Brea, CA, USA), using the ion-selective electrode potentiometer method. Urine specific gravity (USG) was assessed by an experienced laboratory technician with an automatic urinary sediment analyzer (H-800; Dirui, Jilin, China), using the uric dry-chemistry method.

### 2.8. Plasma Biomarker Measurements

Plasma osmolality was measured using standard urine osmolality laboratory methods. The concentrations of plasma electrolytes were tested with the same method measuring the urine electrolyte concentration. Blood glucose (BG) was measured by an experienced laboratory technician using the ion-selective electrode potentiometer method with an automatic biochemical analyzer (AU 5800; Beckman; Brea, CA, USA).

### 2.9. Definition of Hydration Status

The status of hydration was determined on the basis of the osmolality of the urine. Dehydration was defined when the osmolality was more than 800 mOsm/kg. Optimal hydration was defined when the osmolality was less than 500 mOsm/kg [34,35].

### 2.10. Visual Analogue Scales (VAS) for Subjective Sensation

The subjective sensation of thirst in participants was tested by a 10-cm-line [36,37,38]. Participants were instructed to answer the questions by placing a pencil mark anywhere on the line, with extreme answers (“not at all” and “extremely”) at opposite ends of the line. The higher the score, the more thirsty the participant.

### 2.11. Profile of Mood States (POMS)

The POMS questionnaire includes seven subscales and 40 adjectives. Tension, anger, fatigue, depression, confusion, vigor, esteem-related affect, and total mood disturbance (TMD) were measured with the questionnaire. (TMD = (tension + depression + anger + fatigue + confusion) − (vigor + esteem-related affect) + 100) [39]. Each adjective was composed of five levels, from “not at all” to “extremely”. Participants were instructed to choose the levels that corresponded to their mood.

### 2.12. Cognitive Performance (CP)

“Primary cognitive ability” software was used, which was developed by the Institute of Psychology, Chinese Academy Sciences, China. The test consisted of five tasks, including a vocabulary test, similarities test, symbol search test, operation span test, and portrait memory test.

### 2.13. Vocabulary Test

The language comprehension ability was examined by this test. Participants chose the most precise interpretation of the word from five options, which included one most precise option, two precise options, and two unrelated options. There were 36 words in the test. Scores were calculated according to the number of most accurate, accurate, and wrong answers, which were assigned scores of 2, 1, and 0, respectively [40].

### 2.14. Similarities Test

This test assessed verbal comprehension [40]. Participants were asked to find similarities between two words. Further, they had to find the most precise interpretation of the similarities from five options. The test had 32 words, with only one most precise option, two correct options, and two unrelated options for each word, which were assigned 2, 1, and 0 points, respectively. Scores were calculated by totaling the number of most precise, correct, and unrelated answers.

### 2.15. Symbol Search Test

This test measured processing speed [40]. Participants were asked to check two online symbols and five offline symbols in the center of the screen, and they had to find as many similar symbols as possible within the time limit. The duration of the test was 120 s. When they found the right symbol, they received one point.

### 2.16. Operation Span Test

This test evaluated working memory [41]. It consisted of two parts: a mental arithmetic task and remembering Chinese Zodiac Signs task. Participants completed the mental arithmetic task first and then had to remember the Chinese Zodiac Signs that appeared after each problem. After all the mental arithmetic tasks were finished, participants were asked to recall the correct order of the Chinese Zodiac Signs that appeared on the screen for a few seconds after each mental arithmetic task. Every mental arithmetic task had a time limit of a few seconds. When the Chinese Zodiac Signs was recalled in the correct order, the participants would get one point. The scores were only regarded as valid results, until the correct recall rate of the Chinese Zodiac Signs task reached 80%.

### 2.17. Portrait Memory Test

This test assessed episodic memory [42]. The test included six portraits of people. The six portraits appeared in order for a few seconds on the screen, with information about each person’s surname, occupation, and hobby. When portraits were shown for the first time, participants were asked to recall all information about the person on the screen. The portraits were then shown a second time in a different order than the first, and the participants were again asked to recall the information about the person on the screen. When the participants chose the right surname, occupation, and hobby, they received 2 points, 1 point, and 1 point, respectively. Higher scores in this test represented a better ability.

### 2.18. Statistical Analyses

Statistical analyses were performed using SPSS Statistics 20.0 (IBM Corp., Armonk, NY, USA). Quantitative parameters for participants were presented as mean ± standard deviation or median and quartiles, and binary classification data (hydration status) were shown as n (percentage). One-way ANOVA, paired-samples *t* test, or Chi-square test were used to explore differences for normally or non-normally distributed data, respectively. Chi-square tests were used to compare the differences in hydration status. A mixed-model of repeated measurements (TIME × VOLUME) was used to explore the impact of water supplementation on cognitive performance and mood with covariate data such as BMI and blood glucose. The threshold for two-sided significance levels was set at 0.05 (*p* < 0.05) with 95% confidence intervals (95% CI).

## 3. Results

### 3.1. Characteristics of Participants

Sixty-four participants were recruited and completed the study, with half of them male and half female, and the completion rate was 100.0%. The participant characteristics are presented in Table 1. The average age of the participants was 21.0 years. No significant differences were found in the anthropometric measurements, including age, height, weight, BMI, and blood pressure, among the four groups in the dehydration and rehydration tests, respectively (*p* > 0.05).

### 3.2. Temperature and Humidity

The average indoor and outdoor temperatures for day 1 and day 2 were 22.8 °C and 19.3 °C, respectively. The average indoor and outdoor humidity was 72% and 89%, respectively, as shown in Appendix A.

### 3.3. Water Supplementation Effects on Thirst and Hydration Status

As shown in Table 2, comparing rehydration and dehydration tests, a significant interaction between TIME and VOLUME was found for thirst (*p* < 0.05). Significant thirst reductions in WS group 1 and WS group 2 were 1.9 and 1.0, respectively (*p* = 0.003; *p* = 0.042). There was a decrease of 0.3 in WS group 3, but this was not significant (*p* > 0.05). A significant increase of 1.2 was found in the NW group (*p* = 0.039). Comparing the decreases in WS group 1 and WS group 2, no statistically significant differences were found between them (*p* > 0.05).

The results of urine osmolality were consistent with the results of thirst. In urine osmolality, the interaction between TIME and VOLUME was significant (*p* < 0.05). The urine osmolality decreased 461 and 262 mOsm/kg during the rehydration test in WS group 1 and WS group 2 (*p* < 0.001; *p* = 0.004), with a significant increase of 159 mOsm/kg in NW group (*p* < 0.001) compared to the dehydration test. WS group 3 also decreased by 33 mOsm/kg, but not significantly (*p* > 0.05). Comparing WS group 1 with WS group 2, there was no significant difference (*p* > 0.05). Compared to the dehydration test, WS group 1 and WS group 2 had better hydration status at the rehydration test, with nine and five participants changing from dehydrated to optimal hydration (χ^2^ = 16.154, *p* < 0.001; χ^2^ = 11.130, *p* = 0.004). In WS group 3, there was no statistically significant difference in hydration status (*p* > 0.05). In the NW group, the hydration status was worse (χ^2^ = 6.788, *p* = 0.023). In the rehydration test, the hydration status did not differ significantly between WS group 1 and WS group 2, which drank 500 mL and 200 mL of water, respectively (χ^2^ = 5.841, *p* = 0.090).

### 3.4. Water Supplementation Effects on Mood

Results in Table 3 demonstrated that the interactions between TIME and VOLUME for moods related to anger, fatigue, and TMD were different when comparing the dehydration test with rehydration test (all *p* < 0.05). For anger, compared with dehydration test, a significant decrease of 1.7 was found only in WS group 2 (*t* = 2.499, *p* = 0.025), with no significant differences in WS group 3 and the NW group (*t* = −1.031, *p* = 0.319; *t* = 0.212, *p* = 0.835). There was a decrease of 0.7 in WS group 1, but this was not significant (*t* = 1.576, *p* = 0.136). Scores of fatigue and TMD during the rehydration test were significantly lower in WS group 1 and WS group 2 than the scores tested during the dehydration test (*t* = 3.060, *p* = 0.008; *t* = 2.868, *p* = 0.012). Moreover, a significant increase in fatigue was found in the NW group (*t* = −3.257, *p* = 0.005). As for TMD, the increase was borderline significant (*p* = 0.05) in NW group. Comparing WS group 1 and WS group 2, there were no significant differences in the scores of fatigue and TMD in the rehydration test (all *p* > 0.05).

### 3.5. Water Supplementation Effects on Cognitive Performance (CP)

As shown in Table 4, after water supplementation, significant interactions between TIME and VOLUME were found for the operation span test (*p* < 0.05). There were no significant changes for the operation span test in WS group 2, WS group 3, and the NW group (*t* = −1.728, *p* = 0.105; *t* = −0.875, *p* = 0.395; *t* = −0.465, *p* = 0.648), but a significant increase was found in WS group 1 that drank 500 mL of water (*t* = −3.587, *p* = 0.003). There were no significant interactions between TIME and VOLUME for the symbol search test and portrait memory test (*p* > 0.05). In the Symbol Search test, significant increases were found in WS group 1, WS group 2, and WS group 3 (*t* = −3.643, *p* = 0.002; *t* = −3.355, *p* = 0.004; *t* = −2.993, *p* = 0.009), but no significant changes were found in the NW group (*t* = −2.076, *p* = 0.055). In the Portrait memory test, there were no significant differences found for WS group 1, WS group 2, WS group 3, and the NW group (*t* = 1.566, *p* = 0.138; *t* = 1.027, *p* = 0.321; *t* = −0.860, *p* = 0.403; *t* = 1.265, *p* = 0.225).

## 4. Discussion

The current study is, to our knowledge, the first in China to report increased cognitive performance after supplementation with different volumes of water.

It is demonstrated that blood glucose is the major source of energy for the brain and is essential for the normal functioning of the central nervous system. Moreover, there is much evidence that glucose improved various aspects of cognitive performance, such as increasing immediate and delayed recall in episodic memory tasks, and evaluating declarative learning and memory in both young and older healthy adults [43,44]. Therefore, it was also important to measure the levels of blood glucose of the participants during the study in order to explore the effects of water supplementation on cognitive performance merely. Furthermore, the results of the blood glucose did not differ significantly among the participants in our study. Additionally, studies have shown that temperature and humidity affect hydration status, and even cognitive performance [45,46]; therefore, it was necessary to record the indoor and outdoor temperature and humidity of our study area. During the two days of the study, the environment including the temperature and humidity indoors and outdoors was comfortable for participants.

In this study, we observed significant thirst reduction after participants drank 200 and 500 mL water; though, a small volume of water (100 mL) tended to decrease the thirst of participants, but with no significance. Moreover, a significant increase in thirst was found for participants who did not drink water. Therefore, results demonstrated that after water restriction for 12 h, 200 mL of water could alleviate the subjective feelings of thirst, and water supplementation was a sufficient way to prevent dehydration. These results were in line with those of other studies that found similar thirst reductions after drinking larger volumes of water. A study employed among young adults showed that the thirst for those who drank 300 mL of water decreased significantly, while thirst for those who drank 25 mL did not change [33]. Studies implemented in children showed that those who drink 250 or 500 mL water rated themselves as significantly less thirsty [28,47].

Regarding mood, the results of this study demonstrated that water supplementation could alleviate anger, fatigue, and TMD. Consistent with the current findings, a study conducted among young adults after water restriction for 12 h demonstrated that alertness increased after drinking 300 mL of water [48]. Another study with 12 young adults revealed that 1500 mL water supplementation attenuated the decrements of dehydration, including vigor, fatigue, and TMD [23]. A systematic analysis concluded that mood, including feelings of alert or calm, was positively influenced by water supplementation [49]. However, the results of some researches are inconsistent. In a study of 168 children, it was shown that drinking water did not affect the scores of fatigue and confusion, and the alertness, concentration, and energy of 10 young endurance-trained men did not change after fluid replacement [50,51]. Furthermore, a dose–response study confirmed that neither small nor large volumes of water influenced the mood states of children [33].

Our main result was the significant improvement in working memory after water supplementation, in which participants performed better when they received 500 mL of water. Results from this study indicated that only the large amount of water supplementation (500 mL) could improve cognitive performance, not the amounts of 200 and 100 mL. In a study of 52 children aged 9–12 years old, it was revealed that consumption of 750 mL of water affected the working memory [52]. Similarly, in a randomized controlled study, 101 adults were exposed to hot temperatures for four hours, and a significant improvement in working memory was obtained after participants drank 300 mL of water [53]. In another study with 10 male athletes exposed to a high ambient temperature of 32 °C, cognitive performance (executive function) was enhanced after water ingestion [54]. Additionally, one study confirmed the effects of water supplementation of 2.5 L (recommend amount) on working memory in young, healthy women [55]. It was found that drinking 500 mL of water after 12 h dehydration increased the performance in judgement and decision-making tasks for 29 adults [56]. In contrast to the findings mentioned above, some studies did not observe improvement in cognitive performance after water supplementation. One study evaluating cognitive performance in a randomized controlled trial among seven young adults revealed that no beneficial effects of fluid ingestion on cognition tests were obtained [57]. Among well-trained athletes, no differences in choice reaction time were observed after fluid supplementation [51]. In a crossover trial among school children, cognitive performance, including visual attention and short-term memory, were unaffected by water supplementation [58]. The processing speed and episodic memory were also not enhanced by different amounts of water supplementation. Nevertheless, in a study of adults exposed to a temperature of 30 °C for 4 h, it was observed that participants who consumed 300 mL of water had better performance in episodic memory [59]. In addition, in a study of older adult women, it was demonstrated that hydration status and water intake were associated with improved processing speed [60]. Different methods to induce dehydration and various cognitive performance tests may be attributed to the different results of these studies.

Despite the findings from this study were significant, additional studies are necessary to replicate and substantiate these results. Moreover, the mechanisms by which hydration status mediate cognitive performance remain unclear, and further research needs to be conducted. Studies have revealed that exercising in a state of thermal stress and dehydration enhances cognitive performance through increasing the secretion of proteins associated with neurogenesis, such as the serotonin (5-HT), which plays an important role in the expression of brain-derived neurotrophic factor (BDNF) [54,61]. Few studies have used MRI to explore the association between hydration status and brain structure and function. One study showed that the brain volume significantly increased after 1.5 L water consumption, compared to dehydration induced by water restriction for 16 h, among 12 healthy adults [62]. Further, a randomized controlled trial of 13 young adults, investigating the effects of exercise heat stress with and without water supplementation on brain structure and cognitive performance, reported that the brain ventricles decreased and periventricular structures, such as the cerebellum, increased after water replacement compared to dehydration [20]. Another study reported that dehydration and rehydration did not affect brain volume [63]. In children and elderly adults, imaging data suggest that their brains have fewer resources to manage the effects of dehydration, and their cognitive performance was more likely to be impaired compared to young adults [64]. Therefore, future work is needed to determine the mechanism by which water influences cognitive performance and mood among young adults.

This study has some strengths. Firstly, a randomized controlled design was used to reduce bias. Secondly, urine osmolality was measured to monitor the hydration status of participants. Thirdly, it was revealed that water temperature affects cognitive performance and mood; therefore, water temperature was maintained between 30 °C and 40 °C in order to avoid and alleviate gastrointestinal discomfort among the participants. There are also limitations in the current research. Firstly, the impacts of long-term water supplementation on cognitive performance and mood were not explored. Additionally, the mechanisms of changes to cognition and mood after water supplementation, such as brain function and structure and hormone levels, were not investigated in this study.

## 5. Conclusions

In summary, a small amount of water (200 mL) was sufficient to alleviate feelings of thirst, improve hydration status, and attenuate anger, fatigue, and TMD in young adults, but a larger drink (500 mL) appeared to be necessary to improve working memory. The amount of 500 mL was the most appropriate to improve hydration status, cognitive performance, and mood after water restriction for 12 h.

## Figures and Tables

**Figure 1 ijerph-17-07792-f001:**
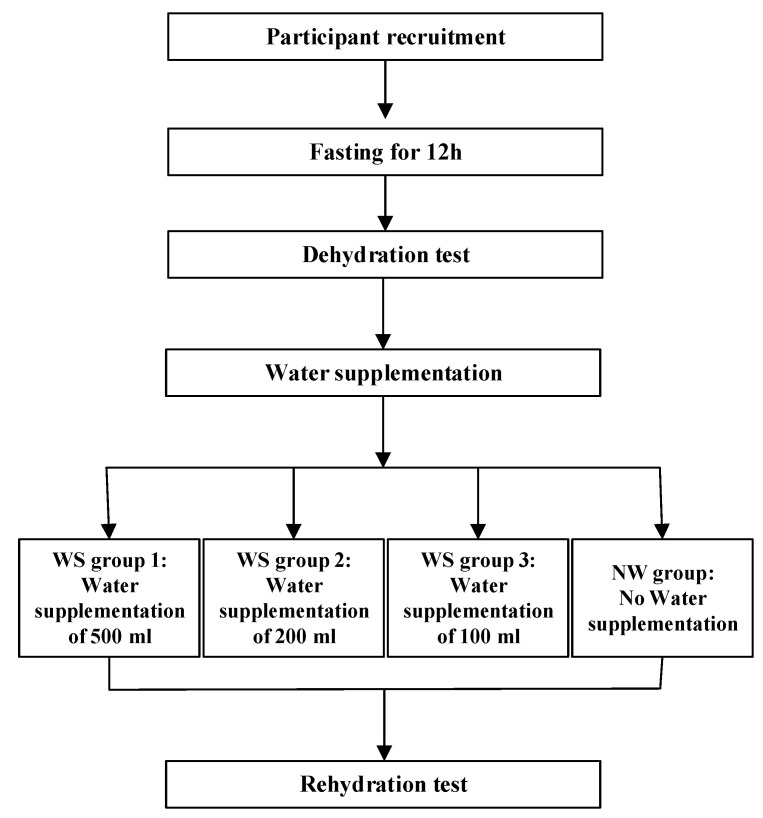
Study Design.

**Figure 2 ijerph-17-07792-f002:**
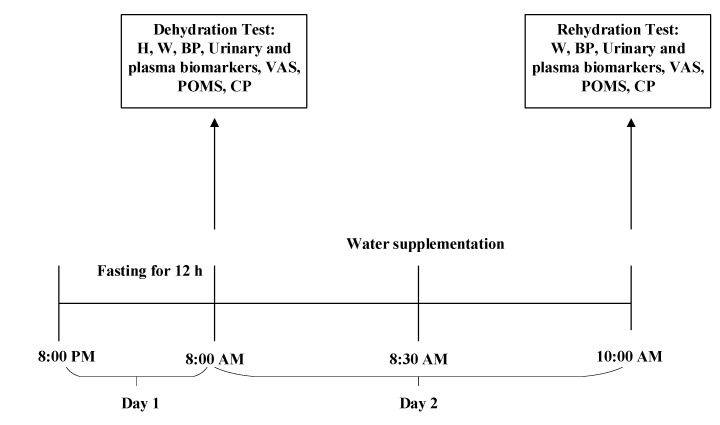
The study procedure. H (Height); W (Weight); BP (Blood Pressure); VAS (Visual Analogue Scales); POMS (Profile of Mood States); CP (Cognitive Performances).

**Table 1 ijerph-17-07792-t001:** Characteristics of the participants.

	Dehydration Test						Rehydration Test					
	WS Group 1 (*n* = 16)	WS Group 2 (*n* = 16)	WS Group 3 (*n* = 16)	NW Group (*n* = 16)	*F*	*p*	WS Group 1 (*n* = 16)	WS Group 2 (*n* = 16)	WS Group 3 (*n* = 16)	NW Group (*n* = 16)	*F*	*p*
Age (year)	21.3 ± 1.4	20.8 ± 1.0	20.9 ± 0.9	21.1 ± 0.8	0.858	0.468	21.3 ± 1.4	20.8 ± 1.0	20.9 ± 0.9	21.1 ± 0.8	0.858	0.468
Height (cm)	166.5 ± 9.3	167.1 ± 8.7	167.6 ± 7.2	162.2 ± 7.5	1.445	0.239	166.5 ± 9.3	167.1 ± 8.7	167.6 ± 7.2	162.2 ± 7.5	1.445	0.239
Weight (kg)	60.5 ± 12.6	63.0 ± 11.5	59.9 ± 5.1	59.2 ± 10.4	0.404	0.750	60.5 ± 12.7	62.9 ± 11.4	59.8 ± 5.2	59.1 ± 10.4	0.421	0.738
BMI (kg/m^2^)	21.6 ± 3.0	22.5 ± 3.4	21.4 ± 1.6	22.4 ± 3.0	0.679	0.568	21.6 ± 3.0	21.3 ± 1.5	22.4 ± 3.0	22.5 ± 3.4	0.661	0.579
Systolic blood pressure (mmHg)	109 ± 11	106 ± 9	103 ± 10	106 ± 11	0.667	0.576	110 ± 10	106 ± 7	104 ± 11	103 ± 11	1.277	0.290
Diastolic blood pressure (mmHg)	65 ± 7	63 ± 8	62 ± 7	66 ± 11	1.013	0.393	65 ± 6	60 ± 7	62 ± 8	63 ± 11	1.271	0.292

Note: Values are mean ± standard deviation (SD). Differences among the four groups were compared using one-way ANOVA in both dehydration and rehydration tests.

**Table 2 ijerph-17-07792-t002:** Thirst, urine, and plasma biomarkers of participants.

	Dehydration Test				*F*	*p*	Rehydration Test				Interaction	
	WS Group 1 (*n* = 16)	WS Group 2 (*n* = 16)	WS Group 3 (*n* = 16)	NW Group (*n* = 16)			WS Group 1 (*n* = 16)	WS Group 2 (*n* = 16)	WS Group 3 (*n* = 16)	NW gRoup (*n* = 16)	*F*	*p*
Thirst	5.5 ± 1.3	5.2 ± 2.2	5.7 ± 2.1	5.1 ± 2.0	0.339	0.797	3.6 ± 1.7 ^#†^	4.2 ± 1.9 ^†^	5.4 ± 2.0	6.3 ± 1.2 ^†^	6.172	0.001
**Plasma Biomarkers**												
Osmolality (mOsm/kg)	290 ± 5	290 ± 6	291 ± 4	291 ± 6	0.190	0.903	288 ± 6	289 ± 7	291 ± 4	290 ± 5	2.314	0.085
Glucose (mmol/L)	4.5 ± 0.3	4.7 ± 0.6	4.5 ± 0.3	4.5 ± 0.4	0.493	0.688	4.7 ± 0.4	4.9 ± 0.6	4.8 ± 0.5	4.8 ± 0.4	0.133	0.940
Na (mmol/L)	140 ± 2	139 ± 3	140 ± 4	140 ± 3	0.560	0.644	138 ± 1	139 ± 3	139 ± 3	139 ± 2	0.632	0.597
K (mmol/L)	4.2 ± 0.4	4.2 ± 0.4	4.3 ± 0.5	4.4 ± 0.4	0.499	0.684	4.3 ± 0.4	4.3 ± 0.5	4.3 ± 0.5	4.4 ± 0.3	0.326	0.807
Cl (mmol/L)	104 ± 2	104 ± 1	104 ± 2	105 ± 2	1.748	0.167	103 ± 2	104 ± 2	104 ± 3	104 ± 1	0.437	0.727
Ca (mmol/L)	2.40 ± 0.07	2.39 ± 0.08	2.40 ± 0.13	2.41 ± 0.10	0.252	0.860	2.42 ± 0.06	2.42 ± 0.07	2.41 ± 0.12	2.42 ± 0.08	0.511	0.676
Phosphorus (mmol/L)	1.31 ± 0.16	1.25 ± 0.15	1.32 ± 0.16	1.29 ± 0.11	0.707	0.552	1.18 ± 0.12	1.24 ± 0.21	1.21 ± 0.16	1.16 ± 0.12	1.675	0.182
Mg (mmol/L)	0.89 ± 0.05	0.87 ± 0.05	0.86 ± 0.05	0.86 ± 0.06	0.850	0.472	0.88 ± 0.04	0.86 ± 0.06	0.85 ± 0.06	0.86 ± 0.06	0.020	0.985
Creatine (mmol/L)	64 ± 15	65 ± 14	66 ± 13	61 ± 12	0.441	0.725	62 ± 15	64 ± 13	66 ± 14	59 ± 11	0.599	0.618
Nitrogen (mmol/L)	4.28 ± 1.19	4.07 ± 0.91	4.46 ± 1.23	4.53 ± 1.14	0.528	0.665	3.80 ± 1.08	3.86 ± 0.76	4.28 ± 1.09	4.34 ± 1.04	1.342	0.269
**Urinary Biomarkers**												
Osmolality (mOsm/kg)	814 ± 221	833 ± 151	820 ± 189	776 ± 134	0.305	0.822	353 ± 200 ^†^	571 ± 237 ^†^	787 ± 231	935 ± 125 ^†^	20.129	<0.001
Volume	270 ± 82	278 ± 97	308 ± 143	298 ± 83	0.444	0.723	334 ± 186 ^#†^	203 ± 90	166 ± 85	145 ± 80	8.430	<0.001
Na (mmol/L)	162 ± 68	184 ± 67	187 ± 59	170 ± 34	0.636	0.594	66 ± 47 ^#†^	110 ± 66 ^†^	167 ± 84	164 ± 64	5.635	0.002
K (mmol/L)	38.6 ± 13.4	43.1 ± 15.0	38.6 ± 12.9	37.8 ± 26.4	0.288	0.834	22.3 ± 10.8 ^#†^	35.8 ± 14.4	58.3 ± 25.8 ^†^	60.1 ± 26.8 ^†^	13.532	<0.001
Cl (mmol/L)	150 ± 44	161 ± 55	168 ± 52	151 ± 30	0.549	0.651	79 ± 50 ^#†^	131 ± 71	201 ± 88	200 ± 67 ^†^	9.178	<0.001
Ca (mmol/L)	1.99 ± 1.20	2.44 ± 1.62	2.98 ± 1.80	2.69 ± 1.16	1.299	0.283	0.88 ± 0.70 ^#†^	1.85 ± 1.56	2.41 ± 1.60	2.93 ± 1.59	2.304	0.086
Phosphorus (mmol/L)	32.73 ± 15.21	37.55 ± 14.26	37.38 ± 14.84	28.84 ± 15.35	1.249	0.300	9.07 ± 7.32 ^#†^	10.89 ± 4.32 ^†^	19.33 ± 7.97 ^†^	23.37 ± 9.30	5.246	0.003
Mg (mmol/L)	2.74 ± 1.06	3.27 ± 1.39	4.10 ± 2.32	2.82 ± 1.06	2.615	0.059	1.25 ± 0.79 ^#†^	2.58 ± 2.02	3.22 ± 1.69	3.00 ± 1.91	1.867	0.145
pH	6.3 ± 0.4	6.5 ± 0.4	6.4 ± 0.3	6.6 ± 0.5	1.924	0.135	6.2 ± 0.4	6.3 ± 0.4	6.3 ± 0.3	6.2 ± 0.4 ^†^	1.677	0.181
USG	1.024 ± 0.006	1.024 ± 0.006	1.023 ± 0.006	1.022 ± 0.006	0.404	0.751	1.013 ± 0.008 ^#†^	1.018 ± 0.006 ^†^	1.023 ± 0.007	1.027 ± 0.006 ^†^	10.822	<0.001
**Hydration Status (%)**												
Dehydrated	9 (56.3)	10 (62.5)	11 (68.8)	7 (43.7)	10.891 ^Φ^	0.092	0 (0.0) ^#^^†^	5 (31.2) ^†^	10 (62.5)	14 (87.5) ^†^	35.359	<0.001
Middle hydration	5 (31.2)	6 (37.5)	2 (12.5)	9 (56.3)			5 (31.2)	3 (18.8)	4 (25.0)	2 (12.5)		
Optimal hydration	2 (12.5)	0 (0.0)	3 (18.7)	0 (0.0)			11 (68.8)	8 (50.0)	2 (12.5)	0 (0.0)		

Note: ^#^, In the rehydration test, the comparison between the four groups was statistically significant; ^†^, there were significant within-group differences comparing the dehydration test with rehydration test. ^Φ^, the statistical value was determined with χ^2^.

**Table 3 ijerph-17-07792-t003:** The POMS of participants.

	Dehydration Test				*F*	*p*	Rehydration Test				Interaction	
	WS Group 1 (*n* = 16)	WS Group 2 (*n* = 16)	WS Group 3 (*n* = 16)	NW Group (*n* = 16)			WS Group 1 (*n* = 16)	WS Group 2 (*n* = 16)	WS Group 3 (*n* = 16)	NW Group (*n* = 16)	*F*	*p*
Tension	3.7 ± 3.0	2.4 ± 2.6	2.2 ± 2.2	2.8 ± 3.6	0.867	0.463	3.3 ± 3.0	2.1 ± 2.3	2.3 ± 2.6	3.2 ± 4.0	0.623	0.603
Anger	1.8 ± 2.8	3.4 ± 5.1	1.4 ± 2.0	2.8 ± 4.4	0.907	0.443	1.1 ± 2.6	1.7 ± 3.6 ^†^	1.8 ± 2.9	2.7 ± 4.5	3.815	0.014
Fatigue	3.8 ± 2.8	3.4 ± 3.5	4.6 ± 4.3	3.4 ± 3.7	0.363	0.780	1.7 ± 1.4 ^#†^	2.3 ± 2.8 ^†^	5.0 ± 4.5	5.1 ± 3.7 ^†^	10.429	<0.001
Depression	2.8 ± 2.8	2.9 ± 3.6	1.2 ± 1.7	2.4 ± 4.0	1.106	0.392	2.0 ± 3.0	1.7 ± 2.7	1.6 ± 2.2	2.7 ± 3.9	2.085	0.112
Confusion	3.6 ± 3.1	2.4 ± 2.7	2.3 ± 1.9	3.3 ± 3.4	0.073	0.974	3.5 ± 3.1	1.8 ± 2.3	2.4 ± 2.2	2.9 ± 3.1	0.402	0.752
Vigor	8.1 ± 4.5	8.2 ± 4.2	8.7 ± 4.6	8.6 ± 4.6	0.836	0.480	9.7 ± 5.2 ^†^	10.0 ± 5.0 ^†^	8.8 ± 4.9	7.1 ± 4.2	2.180	0.100
Esteem-related affect	6.4 ± 3.9	5.7 ± 3.0	5.1 ± 3.4	5.8 ± 3.2	0.400	0.754	6.2 ± 4.0	6.2 ± 3.3	5.6 ± 2.9	5.4 ± 2.8	0.602	0.616
TMD	101.1 ± 13.7	100.6 ± 16.4	97.9 ± 12.6	99.4 ± 17.7	0.138	0.937	95.7 ± 14.3 ^†^	93.4 ± 13.5 ^†^	98.6 ± 13.9	104.1 ± 18.9 ^†^	5.886	0.001

Note: Data are presented as mean ± standard deviation (SD); ^#^, in the rehydration test, the comparison between the four groups was statistically significant; ^†^, there were significant within-group differences comparing the dehydration test with the rehydration test.

**Table 4 ijerph-17-07792-t004:** Cognitive Performance of participants.

	Dehydration Test				*F*	*p*	Rehydration Test				Interaction	
	WS Group 1 (*n* = 16)	WS Group 2 (*n* = 16)	WS Group 3 (*n* = 16)	NW Group (*n* = 16)			WS Group 1 (*n* = 16)	WS Group 2 (*n* = 16)	WS Group 3 (*n* = 16)	NW Group (*n* = 16)	*F*	*p*
Vocabulary test	55 ± 6	58 ± 4	57 ± 3	56 ± 9	0.533	0.662	55 ± 6	58 ± 4	57 ± 3	56 ± 9	-	-
Similarities test	51 ± 7	52 ± 6	51 ± 6	47 ± 8	1.904	0.139	51 ± 7	52 ± 6	51 ± 6	47 ± 8	-	-
Symbol search test	47 ± 5	49 ± 4	48 ± 5	48 ± 5	0.305	0.822	52 ± 6 ^†^	53 ± 6 ^†^	52 ± 7 ^†^	50 ± 5	0.151	0.929
Operation span test	9 ± 2	10 ± 4	11 ± 2	11 ± 3	2.132	0.106	11 ± 2 ^†^	11 ± 3	12 ± 2	11 ± 2	2.816	0.047
Portrait memory test	38 ± 8	37 ± 7	33 ± 10	39 ± 7	1.531	0.216	35 ± 9	35 ± 8	35 ± 12	37 ± 9	1.211	0.314

**Note**: Data are presented as mean ± standard deviation (SD); ^†^, there were significant within-group differences comparing the dehydration test with the rehydration test.

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
