# Peer review of "Different Amounts of Water Supplementation Improved Cognitive Performance and Mood among Young Adults after 12 h Water Restriction in Baoding, China: A Randomized Controlled Trial (RCT)"

_ijerph, 2020, doi:10.3390/ijerph17217792_

Round 1

Reviewer 1 Report

Dear Authors,

Please find below some comments to strengthen the manuscript:

Methods:

I do not understand why some parameters were chosen, e.g. blood glucose? Could Authors refer on this?

On what basis was the number of participants determined? Why the water was served in the temperature of 30-40oC? Is it a typical temperature of drinking water?   

What kind of the paired-samples t test was used? Were the data analyzed for the normality of the distribution?

Results:

Table 1 Please add information about the statistical test used.

Table 2 A large number of explanations/information are placed under the table, which reduces the readability of the results. For greater readability and the comfort of reader, the table should be re-edited.

Table 3 For better readability, please avoid abbreviations in the header. This table is also difficult to follow because of great body of information in the footer.

Table 4 see previous comment.

Discussion:

The limitation of the study was also that it was a single observation.

Authors should emphasize to what extent their results are innovative / what is their practical aspect.

Overall remark: The observed decrease in thirst after water consumption (and no decrease in thirst in the absence of water consumption) seems to be quite an expected result, which does not require confirmation by tests. In my opinion Authors should concentrate more on mood and cognitative results, even skipping the thirst section. 

Author Response

Methods:

1. I do not understand why some parameters were chosen, e.g. blood glucose? Could Authors refer on this?

Response: Thank you for your comments.

It is showed that glucose is the major source of energy for the brain and is essential for the normal functioning of the central nervous system. Moreover, there is much evidence that glucose improved various aspects of cognitive performance, such as increasing immediate and delayed recall in episodic memory tasks, and evaluating declarative learning and memory both in young and older healthy adults.

In order to investigate the effects of water supplementation merely, we measured the blood glucose of the participants, and compared them among the four groups. Furthermore, we used the Mixed model of repeated measurements (TIME × VOLUME) was used to investigate the effect of water supplementation on changes in mood and cognitive performance, whilst covarying age, BMI and blood glucose.

We have made the revision accordingly, and added the references in the Discussion Section of manuscript (Lines 262-269, Page 14).

Following are the added references.

  1. Messier C. 2004. Glucose improvement of memory: a review. Eur J Pharmacol 490: 33–57.
  2. Riby LM, McMurtrie H, Smallwood J, Ballantyne C, Meikle A, Smith E. 2006. The facilitative effects of glucose ingestion on memory retrieval in younger and older adults: is task difficulty or task domain critical? Brit J Nutr 95: 414–420.

2. On what basis was the number of participants determined? Why the water was served in the temperature of 30-40oC? Is it a typical temperature of drinking water?   

Response: Thank you. We have made the revision according to your comments.

The number of participants is determined according to a related study. In the study, the letter cancellation scores of participants in water group before and after water supplementation were 23.34 and 28.81, respectively. The SAS 9.2 (SAS Institute Inc., Cary, NC) was used to calculate the sample size. The power set at 0.8, α set at 0.05 and 10% drop-out rate was taken into account. A total of 64 participants were needed. Eventually, 64 participants were included in the study, including 32 males and 32 females. We have added the information in the Methods Section of the Manuscript (Lines 98-101, Page 3).

In this study, the temperature of the water was maintained between 30℃ and 40℃. In China, people are used to drinking warm or boiled water instead of cold water. Participants would feel uncomfortable if they drank cool or cold water, which may influence the results of the study. Therefore, we maintained the temperature of the water between 30℃ and 40℃, which is suitable for drinking (neither too hot nor too cold), so as to limit any gastrointestinal upset. The water supplied to the participants was kept in three cups. In order to maintain the temperature of the water, the cups were kept in a water bath, and the researchers measured the temperature of the water periodically during the study. We have made the revision accordingly, see Methods Section (Lines 131-136, Page 4).

3. What kind of the paired-samples t test was used? Were the data analyzed for the normality of the distribution?

Response: We have made the revision accordingly.

The paired-samples t test and Chi-square test were used to compared the differences within the same group during dehydration test to rehydration test, if the data were normal distributed or not. We have revised in the Methods Section (Lines 237-239, Page 7).

Results:

Table 1 Please add information about the statistical test used.

Response: We have made the revision accordingly.

The one-way classification ANOVA was used to compare the differences among the four groups in the dehydration test and rehydration test, which was added under Table 1 (Page 8).

Table 2 A large number of explanations/information are placed under the table, which reduces the readability of the results. For greater readability and the comfort of reader, the table should be re-edited.

Response: We have made the revision accordingly.

The table had been re-edited and the explanations such as the statistical values under the Table 2 were removed (Pages 10-11).

Table 3 For better readability, please avoid abbreviations in the header. This table is also difficult to follow because of great body of information in the footer.

Response: We have made the revision accordingly.

The abbreviations in the header of Table 3 was revised and the explanations such as the statistical values under the table were revised accordingly (Pages 9, 12).

Table 4 see previous comment.

Response: We have made the revision accordingly.

The table had been re-edited and the explanations such as the statistical values under the Table 4 were removed in to the Results Section (Pages 9, 13).

Discussion:

The limitation of the study was also that it was a single observation.

Authors should emphasize to what extent their results are innovative / what is their practical aspect.

Response: We have made the revision accordingly.

The results that are innovative were demonstrated in detail in the Discussion Section (Lines 260-261, 347-351, Pages 14,15).

The innovative results of the study were that the study was the first in China to report increased cognitive performance after supplementation with different volumes of water. Moreover, the results of the study showed that a small amount of water (200 ml) was sufficient to alleviate feelings of thirst, improve hydration status, and attenuate anger, fatigue, and TMD in young adults, but a larger drink (500 ml) appeared to be necessary to improve working memory. The amount of 500 ml was the most appropriate to improve hydration status, cognitive performance, and mood after water restriction for 12 h. Based on the above results, people should drink adequate water in order to maintain optimal hydration.

Overall remark: The observed decrease in thirst after water consumption (and no decrease in thirst in the absence of water consumption) seems to be quite an expected result, which does not require confirmation by tests. In my opinion Authors should concentrate more on mood and cognitive results, even skipping the thirst section. 

Response: Thank you for your comments.

In this study, the main aims were to investigate the effects of water supplementation on cognitive performances and mood after water restriction of 12h. The thirst was one of the indicator that maintain the compliance of the participants, and to observe the hydration status they had.

Reviewer 2 Report

I want to commend the efforts of the authors as randomised controlled trials in hydration studies are limited. Well done!

Overall, well written but will benefit from moderate English language editing, especially the tenses. For example, 

Line 44: "Generally, the water was in a dynamic balance, with the intake was equal to the output" could be written as " Generally, water is in a dynamic balance...."

Line 149/150:" After receiving the samples of urine, researchers should send them to the laboratory as soon as possible." This should be written in past tense because it has already happened. i.e "After receiving the samples of urine, researchers sent them to the laboratory as soon as possible. "

The rationale for the study was provided as well as the wider application. 

Methods: The method section is well described and it will be easy to follow mot of the description. However, the study was reported as a randomised, double-blinded experiment but we dont know how participants were randomised or allocated to their respective groups. Who was blinded and how. It will be useful to provide this in your methods.

We know 64 people took part. Was there a difference in the number of people that started and those that completed the study?

Ethics report is good.

Some measures were repeated for accuracy, this is also good. e.g anthropometric measures. 

Temperature and humidity was measured which is useful for hydration status but I couldn't find the discussion on this at all.

Results section: Findings were discussed well and compared to previous research, while highlighting what still needs to be done.

The strength and weaknesses of the study were also discussed

Author Response

I want to commend the efforts of the authors as randomised controlled trials in hydration studies are limited. Well done!

Overall, well written but will benefit from moderate English language editing, especially the tenses.

Response: Thank you for your comments.

The manuscript has been carefully revised by MDPI (English edited 23251), in order to make sure that there were no errors in the English language and grammars throughout the manuscript.

For example, 

Line 44: "Generally, the water was in a dynamic balance, with the intake was equal to the output" could be written as " Generally, water is in a dynamic balance...."

Response: Thank you for your recommendation. We have revised the “Generally, the water was in a dynamic balance, with the intake was equal to the output” into “Generally, water is in a dynamic balance, with intake equal to the output” (Lines 44-45, Page 1).

Line 149/150:" After receiving the samples of urine, researchers should send them to the laboratory as soon as possible." This should be written in past tense because it has already happened. i.e "After receiving the samples of urine, researchers sent them to the laboratory as soon as possible. "

Response: Thank you for your comments. We have made the revision accordingly (Line 168, Page 5).

The rationale for the study was provided as well as the wider application. 

Methods: The method section is well described and it will be easy to follow mot of the description. However, the study was reported as a randomised, double-blinded experiment but we dont know how participants were randomised or allocated to their respective groups. Who was blinded and how. It will be useful to provide this in your methods.

Response: Thank you for your comments.

SPSS software was used to generate random numbers, and then the subjects were randomly divided into the four groups. In order to make the experiment double-blinded, the researchers who distributed the water to the participants and the participants themselves did not know the amount of water they were supplied. Furthermore, the researchers responsible for data analysis did not know the amounts of water supplied to the groups. During analysis, the groups were named as a, b, c, and d instead of WS group 1, WS group 2, WS group 3, and NW group. Moreover, every participant was assigned three opaque cups, and the water supplied to them was divided equally into the three cups. They could not know the amount of water in the cups, even from their weight, so they did not know the amount of water in the cups. Further, they were not allowed to talk with each other about the experiment during the study.

The brief description of the methods of blind was in the Discussion Section, therefore, we moved it into the Methods Section and described it in detail (Lines 137-145, Page 4).

We know 64 people took part. Was there a difference in the number of people that started and those that completed the study?

Response: Thank you for your comments.

Sixty-four participants were recruited and completed the study, with half of them male and half female, and the completion rate was 100.0%. We have revised accordingly in the Results Section (Lines 246-247, Page 7).

Ethics report is good.

Response: Thank you for your comments.

The study protocol was approved by the Peking University Institutional Review Committee (No. IRB00001052-16071). The study was performed according to the guidelines of the Declaration of Helsinki. All participants signed informed consent forms prior to the study (Lines 151-153, Page 5).

Some measures were repeated for accuracy, this is also good. e.g anthropometric measures. 

Response: Thank you for your comments.

In order to the reduce the bias, the height, weight and blood pressure were measured twice (Lines 155-161, Page 5).

Temperature and humidity was measured which is useful for hydration status but I couldn't find the discussion on this at all.  

Response: Thank you for your comments. We have made the revision accordingly.

The Temperature and humidity was measured, and the results in this study showed that the environment of the study place was warm and moist, which was suitable for people. Studies showed that the temperature and humidity affected the hydration status, even the cognitive performances, and therefore, it was necessary to record the information indoors and outdoors of the study places during the study. Furthermore, we wanted to supply more information about the study in order to make the study easy to follow for other researchers.

The manuscript had been revised accordingly; see the Discussion Section (Lines 269-273, Page 14).

Results section: Findings were discussed well and compared to previous research, while highlighting what still needs to be done.

Response: Thank you for your comments.

The strength and weaknesses of the study were also discussed

Response: Thank you for your comments.

Round 2

Reviewer 1 Report

Dear Authors,

Previous comments were adequately addressed in this version of the manuscript.

This manuscript is a resubmission of an earlier submission. The following is a list of the peer review reports and author responses from that submission.